# DUX4 is a common driver of immune evasion and immunotherapy failure in metastatic cancers

Jose Mario Bello Pineda[1,2,3,4], Robert K Bradley[1,2,3]*

[1]Computational Biology Program, Public Health Sciences Division, Fred Hutchinson Cancer Center, Seattle, United States; [2]Basic Sciences Division, Fred Hutchinson Cancer Center, Seattle, United States; [3]Department of Genome Sciences, University of Washington, Seattle, United States; [4]Medical Scientist Training Program, University of Washington, Seattle, United States

**Abstract** Cancer immune evasion contributes to checkpoint immunotherapy failure in many patients with metastatic cancers. The embryonic transcription factor DUX4 was recently characterized as a suppressor of interferon-γ signaling and antigen presentation that is aberrantly expressed in a small subset of primary tumors. Here, we report that *DUX4* expression is a common feature of metastatic tumors, with ~10–50% of advanced bladder, breast, kidney, prostate, and skin cancers expressing *DUX4*. *DUX4* expression is significantly associated with immune cell exclusion and decreased objective response to PD-L1 blockade in a large cohort of urothelial carcinoma patients. *DUX4* expression is a significant predictor of survival even after accounting for tumor mutational burden and other molecular and clinical features in this cohort, with *DUX4* expression associated with a median reduction in survival of over 1 year. Our data motivate future attempts to develop DUX4 as a biomarker and therapeutic target for checkpoint immunotherapy resistance.

## eLife assessment

This study presents a **valuable** finding on the association between DUX4 expression with features of immune evasion in human tissue and clinical outcomes in patients with advanced urothelial cancer. The evidence supporting the claims of the authors is **convincing**, using a range of corroborative statistical techniques. Compared to an earlier version, the quality of the manuscript has been enhanced, for example Figure 5 now illustrates the key features of survival probability estimates over time for patients assigned to with the test or training set.

## Introduction

Immune checkpoint inhibition (ICI) therapy utilizes immunomodulatory monoclonal antibodies to stimulate patient anti-tumor immune responses. Blockade of T cell co-inhibitory receptors, such as CTLA-4 and the PD-1/PD-L1 axis, has achieved major success in the treatment of diverse metastatic cancers compared to first-line chemotherapy (*Doki et al., 2022*; *Hellmann et al., 2019*; *Klein et al., 2020*; *Larkin et al., 2019*; *Motzer et al., 2020*; *Stein et al., 2022*). However, a majority of advanced cancer patients fail to respond to ICI due to de novo or acquired resistance, the mechanistic bases of which remain incompletely understood.

Diverse mechanisms modulate sensitivity and resistance to ICI (*Kalbasi and Ribas, 2020*). These mechanisms include defects in Major Histocompatibility Complex (MHC) class I-mediated antigen presentation due to loss of *B2M* or *HLA* (*Grasso et al., 2018*; *Lee et al., 2020*; *McGranahan et al.,*

*For correspondence:
rbradley@fredhutch.org

**eLife digest** Over time cancer patients can become resistant to traditional treatments such as chemotherapy and radiotherapy. In some cases, this can be counteracted by administering a new type of treatment called immune checkpoint inhibition which harnesses a patient's own immune system to eradicate the tumor. However, a significant proportion of cancers remain resistant, even when these immunotherapy drugs are used. This is potentially caused by tumors reactivating a gene called *DUX4*, which is briefly turned on in the early embryo shortly after fertilization, but suppressed in healthy adults.

Activation of *DUX4* during the early stages of cancer has been shown to remove the cell surface proteins the immune system uses to recognize tumors. However, it remained unclear whether *DUX4* changes the response to immunotherapy in more advanced cancers which have begun to spread and metastasize to other parts of the body.

To investigate, Pineda and Bradley analyzed publicly available sequencing data which revealed the genes turned on and off in patients with different types of cancer. The analysis showed that *DUX4* is reactivated in approximately 10–50% of advanced bladder, breast, kidney, prostate and skin cancers.

Next, Pineda and Bradley studied a cohort of patients with advanced bladder cancer who had been treated with immune checkpoint inhibitors. They found that patients with tumors in which *DUX4* had been turned back on had shorter survival times than patients who had not reactivated the gene.

These results suggest that the activity of *DUX4* could be used to predict which patients with advanced bladder cancer may benefit from immune checkpoint inhibitors. In the future, this work could be extended to see if *DUX4* could be used as a prognostic tool for other types of cancer. Future studies could also investigate if the *DUX4* gene could be a therapeutic target for mitigating resistance to immunotherapy in metastatic cancers.

*2016*; *Sade-Feldman et al., 2017*; *Sucker et al., 2014*; *Wolf et al., 2019*), *PTEN* and *LSD1* inactivation, which sensitizes tumor cells to type I interferon signaling (*Li et al., 2016*; *Peng et al., 2016*; *Sheng et al., 2018*), T cell dysfunction (*Jiang et al., 2018*), presence of specific T cell populations in the tumor microenvironment (*Gide et al., 2019*), and active WNT–β-catenin signaling (*Spranger et al., 2015*). Mitogen-activated protein kinase (MAPK) signaling in *BRAF*-mutated melanomas and CDK4/CDK6 activity have also been implicated in reduced ICI efficacy, and combination treatment with an MAPK/CDK inhibitor improves response to checkpoint blockade (*Ascierto et al., 2019*; *Deng et al., 2018*; *Ebert et al., 2016*; *Goel et al., 2017*; *Jerby-Arnon et al., 2018*; *Ribas et al., 2019*; *Schaer et al., 2018*; *Sullivan et al., 2019*).

Tumor cell-intrinsic interferon-gamma (IFN-γ) signaling is particularly important in anti-tumor immunity. This pathway induces expression of genes involved in MHC class I-mediated antigen processing and presentation, which include genes encoding the TAP1/TAP2 transporters, components of the immunoproteasome, HLA proteins, and B2M (*Alspach et al., 2019*). Thus, suppression of IFN-γ activity promotes tumor immune evasion and decreased CD8+ T cell activation. Indeed, decreased ICI efficacy was observed in patients with tumors harboring inactivating mutations in IFN-γ pathway genes such as *JAK1* and *JAK2* (*Gao et al., 2016*; *Nguyen et al., 2021*; *Sucker et al., 2017*; *Zaretsky et al., 2016*). Similarly, a recent study reported a splicing-augmenting mutation in *JAK3*, linked to decreased *JAK3* expression levels, as a potential mechanism of resistance in a patient with metastatic melanoma treated with anti-PD-1 and anti-CTLA-4 combination therapy (*Newell et al., 2022*).

Some cancers exhibit aberrant expression of embryonic DUX transcription factors. For instance, *DUXB* is expressed in diverse primary malignancies, most notably in testicular germ cell and breast carcinomas (*Preussner et al., 2018*). Recent work from our group and others showed that *DUX4* is expressed in a small subset of primary tumors, where it suppresses tumor cell antigen presentation and response to IFN-γ signaling (*Chew et al., 2019*; *Spens et al., 2023*). We additionally observed signals that *DUX4* expression was associated with reduced survival following response to anti-CTLA-4 or anti-PD-1 in melanoma; however, those analyses relied upon two small cohorts (*n* = 27 or 41 patients), limiting the statistical power of our conclusions.

In its native embryonic context, DUX4 initializes human zygotic genome activation. *DUX4* expression levels peak at the 4-cell stage of the cleavage embryo; *DUX4* is then immediately silenced via

epigenetic repression of the D4Z4 repeat array that contains the *DUX4* gene (*De Iaco et al., 2017*; *Hendrickson et al., 2017*; *Himeda and Jones, 2019*; *Sugie et al., 2020*; *Whiddon et al., 2017*). Aside from select sites of immune privilege such as the testis, *DUX4* remains silenced in adult somatic tissues (*Das and Chadwick, 2016*; *Snider et al., 2010*).

Since *DUX4* expression in cancer cells suppresses MHC class I-mediated antigen presentation (*Chew et al., 2019*), we hypothesized that *DUX4* expression might be particularly common in the setting of metastatic disease (vs. the primary cancers that we studied previously), where immune evasion is particularly important. We therefore analyzed several large cohorts of patients with different metastatic cancers to determine the frequency of *DUX4* expression in advanced disease. We additionally rigorously tested the potential importance of *DUX4* expression for patient response to ICI in a well-powered cohort.

## Results
### *DUX4* is commonly expressed in diverse metastatic cancer types

To assess the prevalence of *DUX4*-expressing human malignancies, we performed a large-scale analysis of publicly available RNA-seq data across diverse cancer types (*Figure 1A*, *Figure 1—figure supplement 1A*). The majority of the cohorts in The Cancer Genome Atlas (TCGA) are most commonly comprised of primary samples and local metastases. We found that *DUX4* expression is a particularly common feature across advanced-stage cancers, with 10–50% of cancer samples (depending upon cancer type) displaying *DUX4* expression levels comparable to or greater than those observed in the early embryo, where expression of the highly stereotyped DUX4-induced gene expression program is observed (*Chew et al., 2019*; *Hendrickson et al., 2017*). A markedly higher proportion of advanced metastatic cancers express *DUX4*—and tend to have higher absolute *DUX4* expression levels—than do their TCGA cancer counterparts (*Figure 1B,C*).

We sought to determine if the *DUX4* transcripts in metastatic cancers express the entire coding sequence or only a portion thereof, as expressed *DUX4* truncations due to genomic rearrangements are frequent oncogenic drivers in particular cancer subtypes, most notably undifferentiated round cell sarcomas (CIC-DUX4 oncoprotein) (*Antonescu et al., 2017*; *Choi et al., 2013*; *Graham et al., 2012*; *Italiano et al., 2012*; *Kawamura-Saito et al., 2006*; *Yoshida et al., 2016*; *Yoshimoto et al., 2009*) and adolescent B-cell acute lymphoblastic leukemia (ALL) (*Lilljebjörn et al., 2016*; *Liu et al., 2016*; *Qian et al., 2017*; *Yasuda et al., 2016*). We aligned RNA-seq reads to the *DUX4* cDNA sequence and examined read coverage over the open reading frame. Resembling the cleavage stage embryo and *DUX4*-expressing primary cancers, *DUX4*-positive metastatic tumors transcribe the full-length coding region. In contrast, B-cell ALL exhibited the expected C-terminal truncation due to *DUX4* fusion with the *IGH* locus (*Figure 1D*).

Since *DUX4* is typically silent in most healthy tissue contexts outside the cleavage-stage embryo (*Das and Chadwick, 2016*; *Snider et al., 2010*), we investigated if artifacts related to sequencing and sample processing could account for the observed high rates of *DUX4* expression in metastatic vs. primary cancers. We were particularly interested in determining whether the method of RNA recovery influenced *DUX4* detection rate, as the analyzed metastatic cohorts frequently relied upon formalin-fixed samples rather than the frozen samples frequently used by TCGA. We took advantage of a cohort of patients with diverse metastatic tumor types for which patient-matched flash-frozen and formalin-fixed metastatic tumor samples were analyzed by RNA-seq (via poly(A)-selection and hybrid probe capture sequencing library preparations, respectively) (*Robinson et al., 2017*). Our re-analysis revealed that *DUX4* expression is readily detectable and quantifiable for both sample and library preparation methods. *DUX4* transcript levels in the majority of the sequenced samples were higher in poly(A)-selected sequencing than were the analogous measurements obtained from hybrid capture (*Figure 1—figure supplement 1B,C*). These data demonstrate that the high rates of *DUX4* expression that we observed across metastatic cancer cohorts reflect true *DUX4* expression rather than technical biases introduced by studying formalin-fixed tissues and are consistent with expression of a polyadenylated *DUX4* transcript in both primary and metastatic cancers.

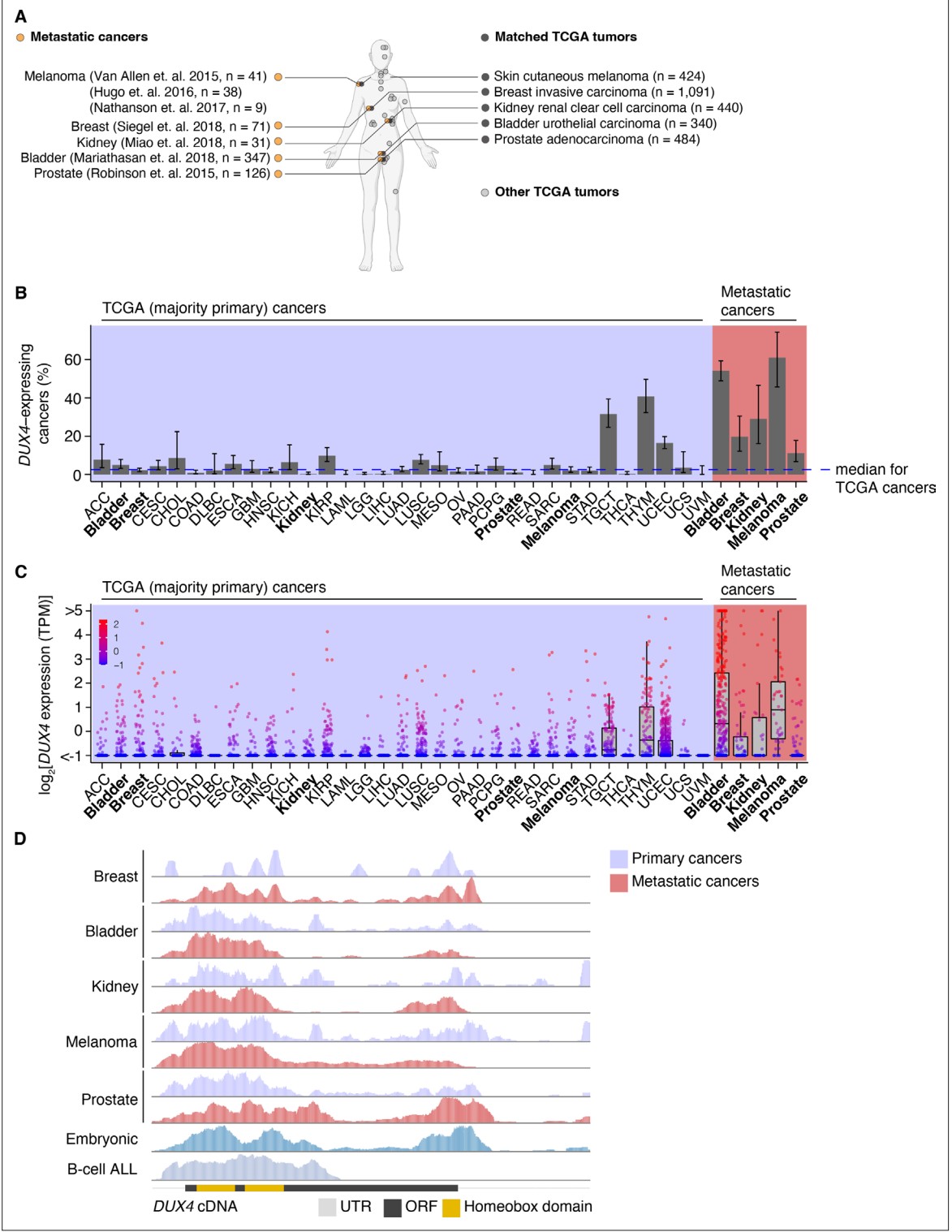

**Figure 1.** *DUX4* is frequently expressed in diverse metastatic cancers. (**A**) Matched The Cancer Genome Atlas (gray, TCGA) and advanced metastatic (orange) cancer datasets analyzed in our study. (**B**) The proportion of *DUX4*-expressing cancers in TCGA (purple shading) and metastatic (red shading) cancers. The blue line indicates the median over TCGA cancer cohorts. The 95% confidence intervals were estimated via a two-sided proportion test. (**C**) *DUX4* expression values (TPM, transcripts per million) in TCGA (purple shading) and advanced metastatic (red shading) cancer cohorts analyzed in our study. (**D**) Representative RNA-seq coverage plots from primary and metastatic cancers for reads mapping to the *DUX4* cDNA. Open reading frame (ORF, black rectangle); UTR (untranslated region, gray line); homeobox domains (yellow rectangles).

The online version of this article includes the following figure supplement(s) for figure 1:

*Figure 1 continued on next page*

*Figure 1 continued*

**Figure supplement 1.** The *DUX4* transcript is likely polyadenylated.

## *DUX4* expression is associated with immune cell exclusion

We next sought to assess the downstream consequences of *DUX4* expression in metastatic cancers. We focused on urothelial cancers for two reasons. First, urothelial cancers exhibited one of the highest frequencies of *DUX4* expression (54% of patients) in any of the five metastatic cancer cohorts that we analyzed, suggesting that DUX4 could be particularly important in that tumor type. Second, pretreatment samples from 347 patients enrolled in the IMvigor210 trial, a phase 2 trial of anti-PD-L1 (atezolizumab) therapy with advanced urothelial carcinoma, were subject to transcriptome profiling by RNA-seq as well as immunohistochemical analysis, enabling us to conduct comprehensive studies of the association between *DUX4* expression, the global transcriptome, and immunophenotypes in a well-powered cohort (*Balar et al., 2017*; *Mariathasan et al., 2018*; *Rosenberg et al., 2016*).

We examined associations between global gene expression profiles and *DUX4* expression in this advanced urothelial carcinoma cohort. We performed differential gene expression analyses on the individuals stratified according to tumor *DUX4* expression status. Gene Ontology (GO) network analyses on the upregulated genes in *DUX4*-positive cancers identified multiple clusters of development-associated terms, consistent with the known role of *DUX4* in early embryogenesis (*Figure 2—figure supplement 1A*; *De Iaco et al., 2017*; *Hendrickson et al., 2017*; *Sugie et al., 2020*; *Whiddon et al., 2017*). In contrast, we found a single network associated with downregulated genes: GO terms corresponding to humoral or cell-mediated immunity (*Figure 2A*). Using an IFN-γ gene signature predictive of response to blockade of the PD-1/PD-L1 axis, we found that *DUX4*-expressing cancers have statistically lower levels of IFN-γ activity (*Figure 2—figure supplement 1B*; *Ayers et al., 2017*). Consistent with IFN-γ suppression, we observed extensive downregulation of genes involved in anti-tumor immunity such as those involved in MHC class I-dependent antigen presentation and T cell activation, checkpoint proteins, and chemokines involved in effector T cell recruitment. *DUX4*-expression was also correlated with suppression of genes critical for MHC class II-mediated antigen presentation, namely: MHC class II isotypes (*HLA–DP/DQ/DR*), *HLA-DM*, and *HLA-DO*, and the invariant chain (*CD74*) (*Roche and Furuta, 2015*). MHC class II gene expression is regulated by the transactivator CIITA via a conserved SXY-module present in the promoter regions of these genes. *CIITA* is induced by IFN-γ and is also conspicuously downregulated in *DUX4*-expressing tumors (*Figure 2B*; *Glimcher and Kara, 1992*; *Masternak et al., 2000*; *Steimle et al., 1993*; *Steimle et al., 1994*). MHC class II-mediated antigen presentation can regulate T cell abundance in the tumor microenvironment and patient response to PD-1 blockade (*Johnson et al., 2020*). These analyses suggest that *DUX4* expression in the metastatic context induces an immunosuppressive gene expression program, concordant with its established function in inhibiting JAK–STAT signaling in primary cancers (*Chew et al., 2019*).

We hypothesized that *DUX4* expression in these cancers will generate related transcriptomic signals consistent with CD8[+] T cell exclusion from the tumor. We assessed this using an effector CD8[+] T cell transcriptomic signature developed from initial studies of the IMvigor210 phase 2 trial (*Balar et al., 2017*; *Rosenberg et al., 2016*). *DUX4*-expressing cancers had lower measures of the gene signature, consistent with decreased CD8[+] T cell infiltration into the tumor (*Figure 2C*). We also investigated the possible effects of *DUX4* expression on the exclusion of other immune cell types using gene signatures developed from TCGA (*Danaher et al., 2017*). In these analyses, we recapitulated the observation of lower CD8[+] T cell signature associated with *DUX4* positivity (*Figure 2—figure supplement 1C*). In addition, we observed patterns consistent with widespread immune cell exclusion from the tumor microenvironment (*Figure 2—figure supplement 1D*).

Defects in chemokine signaling could partially account for the observed *DUX4*-associated decrease in immune gene signature measurements. To test this hypothesis, we examined expression of chemokines involved in immune cell recruitment. In *DUX4*-expressing cancers, we observed lower mRNA levels of *CXCL9* and *CXCL10*, chemokines which recruit T cells to the tumor site (*Figure 2D,E*; *Nagarsheth et al., 2017*). Utilizing a chemokine signature associated with host immune response to solid tumors, we observed that *DUX4* expression was correlated with broad reduction in the expression of chemokine signaling genes, beyond T cell-associated signals (*Figure 2—figure supplement 1E*; *Coppola et al., 2011*).

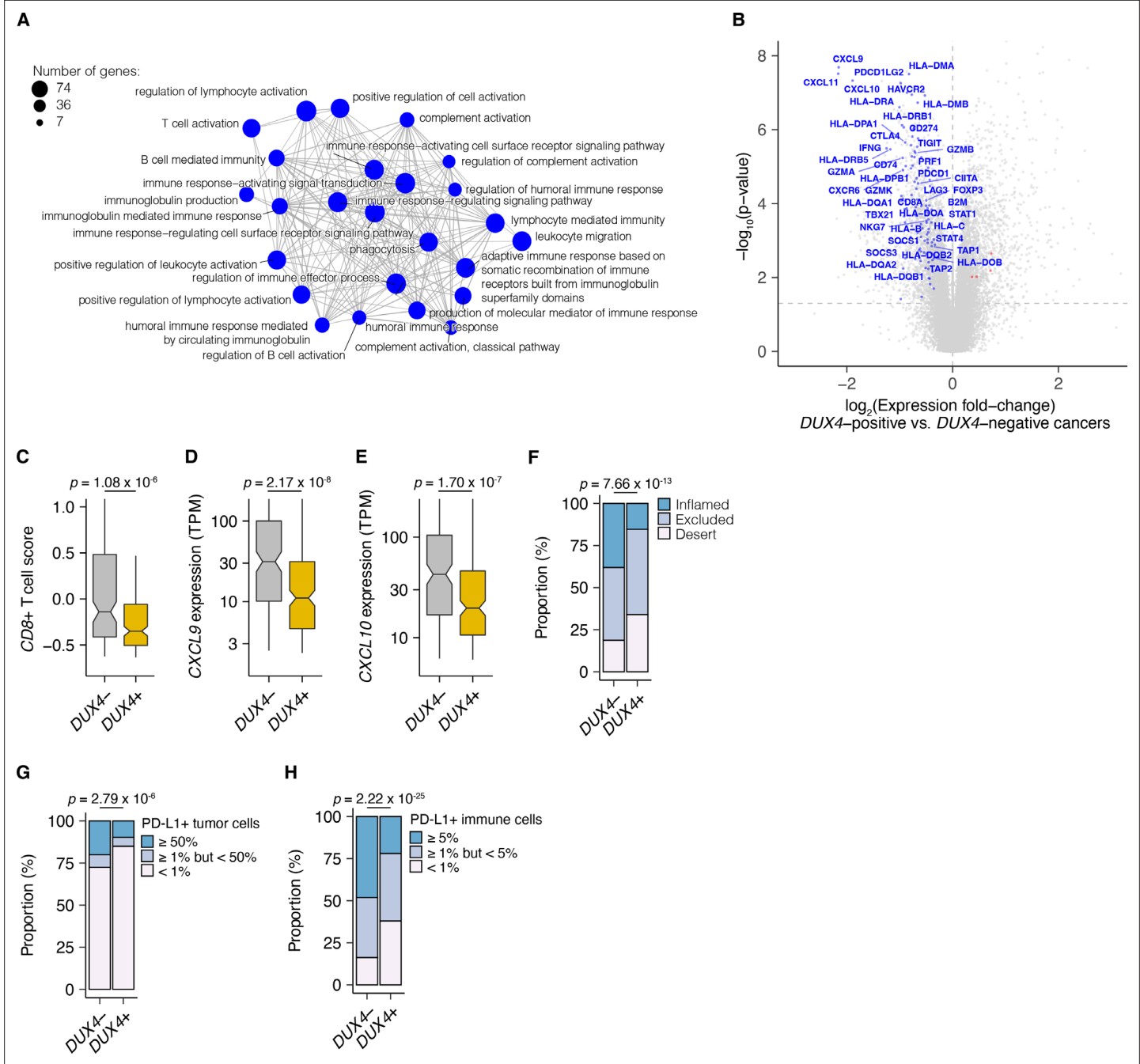

**Figure 2.** *DUX4* expression in advanced cancers is associated with signatures of host anti-tumor immunity inhibition. (**A**) Gene Ontology (GO) enrichment network analysis of *DUX4*-downregulated genes. Differentially expressed genes were identified from the comparison of advanced urothelial carcinoma tumors with high (>1 TPM) vs. low (≤1 TPM) *DUX4* expression. The nodes and node sizes correspond to significantly enriched GO terms (Benjamini–Hochberg-adjusted p-value <0.05) and the number of *DUX4*-downregulated genes in each, respectively. The edges connecting nodes indicate shared genes. (**B**) Downregulated (blue) and upregulated (red) anti-tumor immunity genes in tumors with *DUX4*-positive (>1 TPM) vs. -negative (≤1 TPM) advanced urothelial carcinomas. (**C**) Effector CD8[+] T cell score, defined as the mean of the *z*-score normalized gene expression values in the signature (***Mariathasan et al., 2018***) for *DUX4*+/− tumors. The p-value was estimated via a Mann–Whitney *U* test. (**D**) *CXCL9* expression for *DUX4*+/− tumors. The p-value was estimated via a Mann–Whitney *U* test. (**E**) As in (**D**), but illustrating *CXCL10* expression. (**F**) Proportion of immune phenotypes in *DUX4*+/− cancers. The phenotypes were based on the CD8[+] T cell abundance and degree of tumor infiltration determined by anti-CD8 staining of tumor formalin-fixed paraffin-embedded (FFPE) sections in the original study (***Mariathasan et al., 2018***). The p-value was estimated via a multinomial proportion test. (**G**) PD-L1 expression on tumor cells stratified by *DUX4* expression status measured by immunohistochemistry in the original study. The samples were categorized based on the percentage of PD-L1-positive tumor cells. The p-value was estimated via a multinomial proportion test. (**H**) As in (**G**), but PD-L1 staining on tumor-infiltrating immune cells (lymphocytes, macrophages, and dendritic cells) is represented.

*Figure 2 continued on next page*

*Figure 2 continued*

The online version of this article includes the following figure supplement(s) for figure 2:

**Figure supplement 1.** *DUX4* positivity is correlated with an embryonic gene expression signature, downregulation of interferon-gamma (IFN-γ) signaling, and exclusion of diverse immune cell types.

We directly assessed the correlation of *DUX4* expression to immune cell exclusion by examining CD8+ T cell abundance in the tumor microenvironment, measured by immunohistochemistry (IHC) on formalin-fixed paraffin-embedded patient tumor sections. We verified that *DUX4* expression in the advanced urothelial carcinoma tumors was associated with an immune exclusion phenotype: a higher proportion of *DUX4*+ tumors exhibit either an immune-excluded or immune-desert phenotype compared to malignancies where *DUX4* is silent (**Figure 2F, Figure 2—figure supplement 1F**). We similarly examined the correlation of *DUX4* expression status with PD-L1 levels in the tumor and immune compartments quantified via IHC. We determined that *DUX4* expression was associated with a significant decrease in PD-L1 levels on both tumor and host immune cells, consistent with *DUX4*-induced suppression of IFN-γ signaling (**Figure 2G,H, Figure 2—figure supplement 1G,H**). PD-L1 expression on immune cells such as dendritic cells and macrophages modulate anti-tumor immune suppression and response to ICI in in vivo mouse models (**Lau et al., 2017; Lin et al., 2018; Noguchi et al., 2017**). Importantly, PD-L1 levels on immune cells are correlated with response to ICI in clinical trials (**Powles et al., 2014; Rosenberg et al., 2016**).

## DUX4 expression is correlated with poor response to ICI in advanced urothelial carcinoma

Given the correlation between cancer *DUX4* expression and signatures of anti-tumor immune response suppression, we sought to understand if *DUX4* expression in patient tumors was associated with accompanying changes to overall survival during PD-L1 inhibition. *DUX4* expression was associated with a significant decrease in objective response rates, assessed using the Response Evaluation Criteria in Solid Tumors (RECIST) (**Figure 3A**). As expected, higher tumor mutational burden (TMB) was linked to improved survival outcomes in this cohort (**Figure 3B**). Interestingly, we found that *DUX4* expression was correlated with statistically lower survival rates in this cohort after crudely adjusting

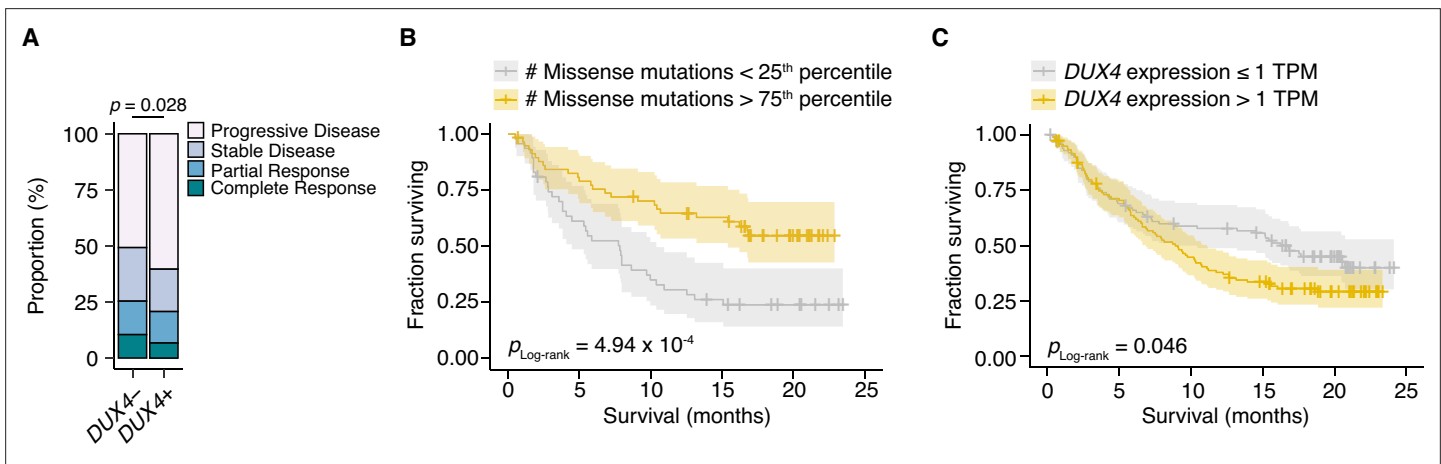

**Figure 3.** *DUX4* positivity is associated with decreased response to immune checkpoint inhibition. (**A**) The proportion of clinical response classifications (RECIST, Response Evaluation Criteria in Solid Tumors) in *DUX4*-positive (*DUX4*+, >1 TPM) or -negative (*DUX4*−, ≤1 TPM) advanced urothelial carcinoma patients. RECIST categories were assigned in the original study (**Mariathasan et al., 2018**). The p-value was estimated via a multinomial proportion test. (**B**) Kaplan–Meier (KM) estimates of overall survival for the patients in (**A**) stratified by tumor mutational burden (TMB, number of missense mutations). The estimated survival functions (solid lines), censored events (crosses), and 95% confidence intervals (transparent ribbons) for the patients in the top and bottom TMB quartiles are plotted. The p-value was estimated via a log-rank test. (**C**) As in (**B**), but patients are stratified by *DUX4* expression. To control for possible confounding by TMB, the quartile of patients with the lowest TMB was excluded.

The online version of this article includes the following figure supplement(s) for figure 3:

**Figure supplement 1.** *DUX4* expression status stratifies patients according to survival.

for the effects of TMB by removing the bottom quartile of patients, those with the lowest number of missense mutations in their tumors (*Figure 3C*, *Figure 3—figure supplement 1A*). Those results motivated us to more carefully control for the effects of other patient covariates in order to better clarify the effects of *DUX4* expression on survival.

## Risk assignments are improved with *DUX4* expression

We next determined whether *DUX4* expression was a significant predictor of survival for ICI-treated patients after controlling for TMB and other potentially relevant variables in a statistically rigorous manner. We used Cox Proportional Hazards (PH) regression to quantify the effects of multiple clinical, demographic, and molecular features on risk of death during ICI. For these and subsequent analyses, we elected to define *DUX4*-negative samples as those with *DUX4* expression levels <0.25 TPM. This scheme excludes 126 patients but presumably produces more reliable categorizations, avoiding potential misclassifications due to loss of sensitivity of bulk RNA-seq at very low expression levels (*Mortazavi et al., 2008*). In the context of multivariate Cox PH regression, which controls for the confounding effects of all other covariates simultaneously, we observed that TMB was positively associated with survival [hazard ratio (HR) = 0.14], as expected. Conversely, *DUX4* expression, Eastern Cooperative Oncology Group Performance Status (ECOG PS) >0, and previous administration of platinum chemotherapy were correlated with increased risk (or shorter survival), while other features that have previously been reported as associated with reduced survival e.g., *TGFB1* expression (*Mariathasan et al., 2018*) did not remain significant after controlling for TMB and other variables. In particular, *DUX4* positivity was associated with dramatically worse survival, with a 3.2-fold increase in risk of death at any point in time compared to *DUX4*-negative status (*Figure 4A*, *Supplementary file 1*, *Supplementary file 2*).

We next investigated if *DUX4* expression status carried added value as a predictor over routinely collected clinical and molecular information. We focused on the variables with significant HRs under both the univariate and multivariate regression settings: *DUX4* expression status, TMB, ECOG PS, and history of platinum chemotherapy. We employed goodness-of-fit measurements, which compare the observed data to expectations from Cox PH models created using various combinations of the covariates. In these analyses, we observed a quantifiable improvement in data-model congruence with the addition of *DUX4* expression status (*Figure 4B*, *Figure 4—figure supplement 1A,B*). Additionally, we measured statistically significant differences in the likelihoods of the reduced models (without *DUX4* expression as a predictor) when compared to the full model (employs all covariates) (*Supplementary file 3*). Taken together, these analyses indicate that *DUX4* expression status is an informative predictor of risk under ICI treatment.

We evaluated the utility of *DUX4* expression status for pretreatment risk assignment in predicting patient response to ICI. We trained full and reduced Cox PH models on randomly sampled patients (training set, 70% of the cohort) and quantified their respective risk scores. A reference risk score per model was computed as the median score across the training set and was used to ascribe patients into low- vs. high-risk groups. Using these models, we quantified risk scores on the individuals excluded from model construction (test set, 30% of the patients), and similarly assigned patients into low- or high-risk groups based on the training set reference score. By empirically quantifying survival of the two risk groups using KM (Kaplan–Meier) estimation, we found that the full model stratifies patients in an informative manner, appropriately discriminating patients with longer vs. shorter survival times (*Figure 4C*, *Figure 4—figure supplement 1C,D*). Furthermore, the addition of *DUX4* expression status improves model performance as illustrated by the time-dependent Brier score, a measure of survival prediction accuracy at specific timepoints (*Figure 4—figure supplement 1E*).

## *DUX4* expression impedes response to ICI after controlling for other clinical characteristics

We used a Random Survival Forest (RSF) model to quantify the effect of *DUX4* expression on survival in ICI-treated advanced urothelial carcinoma patients (*Ishwaran et al., 2008*). The RSF is a machine learning ensemble, an extension of the Random Forest algorithm for right-censored data (*Breiman, 2001*). It can provide accurate estimates of risk and survival probability at definite times by aggregating predictions from a multitude of base learners (survival trees) (*Ishwaran et al., 2008*). RSFs have been successfully used to study time-to-event problems in medicine, including measurement of

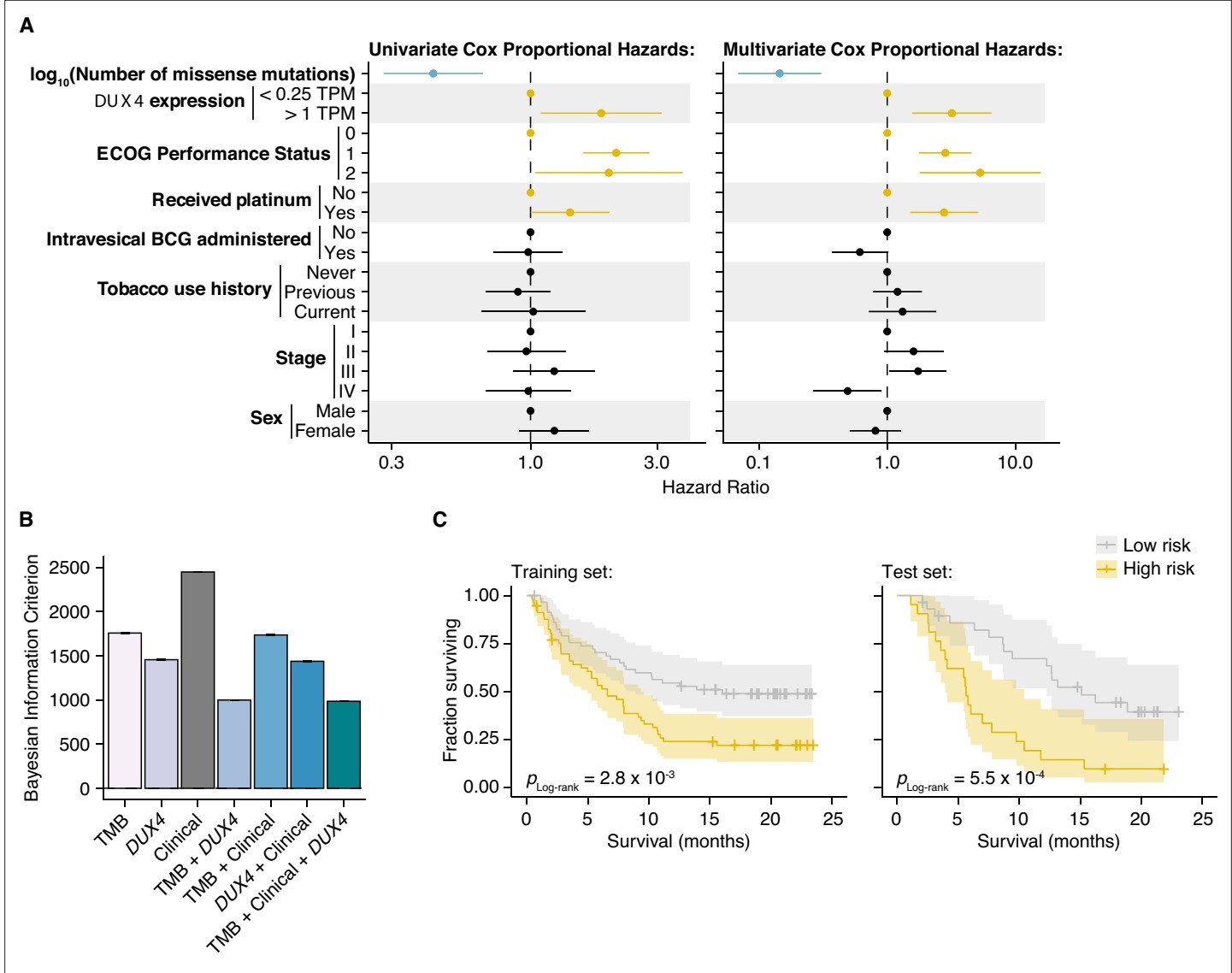

**Figure 4.** *DUX4* expression status affects clinical response after controlling for other genetic and clinical variables. (**A**) Hazard ratios (HRs) and 95% confidence intervals for the variables included in univariate (left) or multivariate (right) Cox Proportional Hazards (PH) regression. For categorical variables, the reference groups are indicated by points at HR = 1. Statistically significant predictors that are associated with increased (orange) or decreased (blue) risk in both the univariate and multivariate contexts are highlighted. ECOG (Eastern Cooperative Oncology Group); BCG (Bacillus Calmette–Guerin). (**B**) Bayesian information criterion (BIC) measurements for goodness of fit for the full (tumor mutational burden [TMB], Clinical, DUX4 expression) vs. reduced Cox PH models, where lower values indicate better fit. The bootstrapped BIC mean and the 95% confidence interval are illustrated. Clinical (ECOG Performance Status and Platinum treatment history). (**C**) Kaplan–Meier (KM) estimates of overall survival, 95% confidence interval (transparent ribbon), and censored events (crosses) for low-risk (solid gray line) and high-risk (solid orange line) patients in the training (left) and test (right) sets. Risk group assignments were based on risk scores estimated by the full Cox PH model. p-values were estimated via a log-rank test.

The online version of this article includes the following figure supplement(s) for figure 4:

**Figure supplement 1.** Cox Proportional Hazards regression models containing *DUX4* expression status as a predictor have a better fit to the data.

variable importance (*Dietrich et al., 2016*; *Hsich et al., 2019*; *Ishwaran et al., 2009*; *O'Brien et al., 2021*; *Semeraro et al., 2011*). We utilized the RSF model to address potential limitations of our Cox PH analyses. First, the RSF model is fully non-parametric and as such does not operate under the Cox PH assumptions: a constant relative hazard between strata over time (PH), a linear relationship between the predictors and the log hazard, and the unspecified baseline hazard function. Second, the RSF model can compute estimates of absolute risk and survival probability over time independent of a reference, unlike relative risk models such as Cox PH (*Ishwaran et al., 2008*).

We used all available molecular, clinical, and demographic covariates to grow an RSF. We randomly selected 70% of the patients to grow the forest, with the resulting model having an out-of-bag (OOB) error of 38.4%. The OOB error stabilizes with increasing number of trees and converges to the leave-one-out cross-validation error estimate. Thus, OOB error is characterized as an unbiased estimate of the model's true prediction error (**Breiman, 2001**; **Hastie et al., 2009**). In some instances, the OOB error provides overestimates and some reports have recommended treating it as an upper bound (**Bylander, 2002**; **Janitza and Hornung, 2018**; **Mitchell, 2011**). Thus, we measured the RSF model's test error using a holdout set (the remaining 30% of the cohort) excluded from training. The RSF model recorded a test error of 32.6% illustrating an appropriate fit (**Figure 5—figure supplement 1A**). Our error measurements are comparable to **Ishwaran et al., 2008**, suggesting our model can be used for inference purposes. Furthermore, the time-dependent Brier score of the RSF model on the training and test sets confirms informative survival prediction (**Figure 5—figure supplement 1B**).

The RSF model predicted worse survival outcomes in patients with *DUX4*-expressing cancers compared to their *DUX4*-silent counterparts. These predictions were mirrored in the test dataset, illustrating robustness of the model (**Figure 5A**). Using time-dependent receiver operating characteristic (ROC) curve analyses, we identified the time range for which the RSF predictive performance is statistically divergent from random guessing: approximately 6–20 months (**Figure 5—figure supplement 1C**). In this window, we observed significant survival differences between patients with *DUX4*+ and *DUX4*− tumors. We highlighted the model's performance at predicting 1- and 1.5-year survival, typical timepoints of clinical interest. For these times, the RSF appropriately discriminates patient death and survival (**Figure 5—figure supplement 1D**). Examining the absolute effects of *DUX4* expression on survival, the RSF model predicted an approximately 20% decrease in both 1- and 1.5-year survival probabilities in patients with *DUX4*-expressing cancers (**Figure 5B**). Importantly, RSF survival predictions conform closely with the empirical survival estimates obtained via the Kaplan–Meier model (**Figure 5C**).

We sought to determine the importance of *DUX4* expression status relative to the other covariates in the RSF model. We measured feature importance using estimated Shapley values, which quantify the marginal contribution of each variable to the RSF prediction (**Lundberg and Lee, 2017**; **Maksymiuk et al., 2020**; **Shapley, 1953**; **Štrumbelj and Kononenko, 2014**). Specifically, Shapley values measure variable contributions to predictions at the level of each patient. Contributions to the overall performance of the RSF model can be assessed by examining the aggregated summary: the average of the absolute Shapley values for a predictor across the patient cohort. We estimated Shapley values associated with predicting ensemble mortality, the RSF risk estimate. In these analyses, ECOG PS had the largest contribution, followed by TMB and *DUX4* expression (**Figure 5—figure supplement 1E**). We validated these feature rankings using two independent metrics. The first metric was permutation importance, which quantifies the change in prediction error associated with permutation of a variable's data; important covariates will record large deviations from the original predictions (**Breiman, 2001**; **Ishwaran, 2007**). The second measure employed was minimal depth, a measure of the variable-node-to-root-node distance within the survival trees of the RSF; important variables tend to have smaller minimal depth values as they are typically used for earlier decision splits (**Ishwaran et al., 2010**; **Ishwaran et al., 2011**). Feature contributions measured using permutation importance and minimal depth were consistent with the Shapley-based assignments, notably identifying *DUX4* expression as an important contributor to patient survival outcomes (**Figure 5—figure supplement 1E**). We investigated time-dependent changes in variable importance by estimating Shapley values associated with predicting survival probability at distinct timepoints along the observation window. Interestingly, we observed the strong dependence on ECOG PS for predicting survival at early timepoints under this paradigm. The importance of *DUX4* expression for survival prediction is most prominent at later times (**Figure 5D**). Altogether, we found that diverse variable importance measures converge on identifying *DUX4* as a major contributor to patient survival prediction.

We sought to quantify the effect of *DUX4* expression on survival predictions after controlling for the effects of the other covariates. With Shapley dependence plots, which allows visualization of the marginal effects of a variable on the predicted outcome, we measured the expected negative correlation between TMB and mortality (**Figure 5—figure supplement 1F**; **Lundberg et al., 2020**). We performed a similar dependence analysis on *DUX4* expression and observed a clear separation of positive and negative Shapley values based on *DUX4*-positive and -negative status, respectively. These

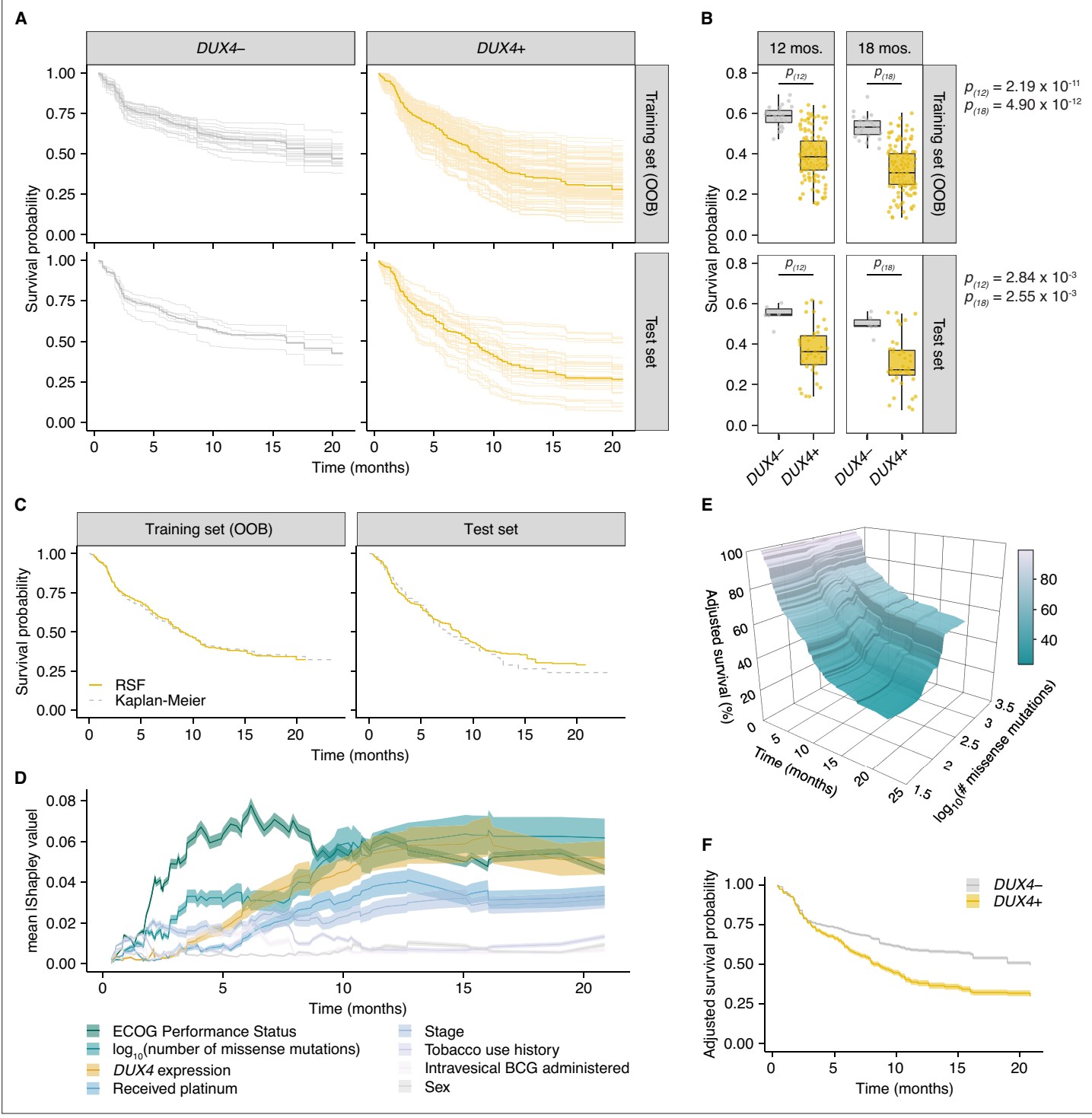

**Figure 5.** *DUX4* expression is associated with decreased overall survival in the context of immune checkpoint inhibition. (**A**) Random Survival Forest (RSF) predicted overall survival for patients with either *DUX4*-positive or -negative tumors in the training and test sets. Out-of-bag (OOB) survival predictions are shown for the patients in the training set. Survival predictions for individual patients (thin lines) and the median survival function across the cohort (thick line) are represented. *DUX4*− (<0.25 TPM); *DUX4*+ (>1 TPM). (**B**) Training (OOB) and test set survival probability predictions for patients with *DUX4*+/− tumors at 12 and 18 months. The p-values were estimated using a two-sided Mann–Whitney *U* test. (**C**) Survival probability for patients in the training and test sets. Survival functions corresponding to the median RSF prediction (solid orange line) and the Kaplan–Meier estimate (dashed gray) are displayed. (**D**) Feature importance for variables used in the RSF model. The average absolute estimated Shapley values (solid lines) are shown, associated with predicting survival probability at particular times. The 95% confidence interval of the mean (transparent ribbon) is plotted.(**E**) Surface plot showing adjusted (marginal) survival probability, measured via partial dependence, as a function of tumor mutational burden (TMB, number of

*Figure 5 continued on next page*

*Figure 5 continued*

missense mutations) and time. Each point on the surface corresponds to the mean survival prediction (at the respective timepoint) after TMB is fixed to the respective value for all patients. (**F**) Partial plot showing adjusted survival probability as a function of *DUX4* expression status. The median survival probability (solid lines) and the 95% confidence interval (transparent ribbon) after *DUX4* expression status is fixed to the indicated value for all patients are plotted.

The online version of this article includes the following figure supplement(s) for figure 5:

**Figure supplement 1.** A Random Survival Forest (RSF) model quantifies the effect of *DUX4* status on overall survival probability in the context of immune checkpoint inhibition.

results signify an increase in predicted risk of death associated with *DUX4* expression (*Figure 5—figure supplement 1G*). Shapley values are scaled, variable attributions which add up to the difference between the prediction for an individual and the average prediction across the entire cohort and thus require transformation to get absolute measures of risk. To quantify the effects of TMB and *DUX4* expression in the appropriate risk units (mortality, expected number of deaths), we utilized partial dependence as an alternative way to represent mortality predictions as a function of these variables, marginalized over the other predictors in the data (*Friedman, 2001*). Specifically, the average model predictions across the individuals in the cohort are calculated over the unique predictor values. The marginal effects of TMB and *DUX4* expression measured via partial and Shapley dependence analyses show strong concordance. Comparison of the patients with the lowest vs. highest TMB revealed an approximately 67% difference in mortality. On the other hand, we measured an approximately 53% increase in mortality associated with *DUX4* positivity (*Figure 5—figure supplement 1H,I*). We then extended the partial dependence analyses to survival probability predictions over time. In this paradigm, we similarly observed that higher TMB was correlated with increased survival probability, more pronounced at later times (*Figure 5E*). *DUX4* expression was correlated with poorer survival outcomes, with a 1- and 1.5-year survival difference of 20.7% and 19.2% between patients with *DUX4+* and *DUX4−* tumors, respectively. Strikingly, our analyses measure a difference of at least 12.5 months in median survival between the *DUX4+* and *DUX4−* strata (*Figure 5F*). Overall, our analyses demonstrate a significant and robust decrease in survival attributable to *DUX4* expression in advanced cancers.

## Discussion

*DUX4* expression is a common feature of metastasis and may be an important driver of immune evasion. While the mechanism governing *DUX4* de-repression in cancer remains to be elucidated, we show that *DUX4* expression in the metastatic context is associated with reduced anti-tumor immunity, mirroring previous observations in primary cancers and cancer cell line models (*Chew et al., 2019*), and is correlated with decreased patient survival under ICI treatment.

The prognostic value of IFN-γ activity (*Ayers et al., 2017*; *Grasso et al., 2020*; *Newell et al., 2022*) and its non-redundancy relative to TMB in terms of influencing ICI response is widely appreciated (*Cristescu et al., 2018*; *Newell et al., 2022*; *Rozeman et al., 2021*). For example, patients with advanced melanoma that is nonresponsive to anti-CTLA-4 or anti-PD-1/PD-L1 therapy have higher frequencies of genetic alterations associated with IFN-γ signaling defects compared to responsive patients (*Gao et al., 2016*; *Nguyen et al., 2021*; *Sucker et al., 2017*). DUX4 has been implicated in modifying IFN-γ activity through direct binding and inhibition of STAT1 via its C-terminal domain (*Chew et al., 2019*; *Spens et al., 2023*). Our sequence analyses show that *DUX4* transcripts in the metastatic context contain the full-length coding region, suggestive of an intact capability as a STAT1 suppressor. *DUX4*'s ubiquitous expression across metastatic cancers and our controlled survival analyses emphasize *DUX4* as an underappreciated contributor to ICI resistance.

Our RSF model allowed us to interrogate changes in variable importance over time. For instance, the contribution of *DUX4* expression to survival prediction is most prominent at later timepoints, suggesting principal effects on long-term survival. Intriguingly, these analyses revealed the outsize influence of ECOG PS, a measure of patient functional status, on survival at early timepoints relative to other patient covariates. ECOG PS negatively impacts patient survival during ICI therapy, inferred from our multivariate Cox PH analysis and from findings of the IMvigor210 clinical trial: patients with ECOG PS = 2 (*n* = 24) had a median overall survival of 8.1 months, lower than the subgroup with ECOG PS

<2 (*n* = 35) whose median survival was not reached during the observation period (*Balar et al., 2017*). Other studies have similarly reported poorer outcomes associated with ICI treatment in patients with high ECOG PS (*Chalker et al., 2022*; *Krishnan et al., 2022*; *Cristescu et al., 2018*; *Sehgal et al., 2021*). Altogether, these results possibly indicate the existence of co-occurring conditions in patients with higher degrees of disability, predisposing them to adverse effects associated with ICI treatment— comorbidities whose effects presumably manifest shortly after therapy commencement. Our data underscore the utility of time-dependent approaches in identifying covariate-linked survival effects which may not be apparent in a summary computed over the entire time period.

Our results may have broad implications for ICI treatment. First, *DUX4* expression may promote patient resistance in a wide array of ICI modalities. Our previous work showed that *DUX4* expression is associated with resistance to anti-CTLA-4 and anti-PD-1 therapies (*Chew et al., 2019*). In our current study, we comprehensively demonstrate that *DUX4* modulates patient response to PD-L1 blockade. We also report that *DUX4* expression in metastasis is correlated with downregulation of *TIGIT* (*Zhang et al., 2018*) and other immune checkpoints whose interception are currently under clinical investigation: *HAVCR2/TIM3* (NCT02608268; *Dixon et al., 2021*) and *LAG3* (NCT02658981; *Amaria et al., 2022*; *Tawbi et al., 2022*). Second, the pervasive expression of *DUX4* in all the metastatic cohorts we examined exhibits its potential as a pan-cancer biomarker. We show that binary categorization of patients according to *DUX4* expression status was sufficient to stratify patients according to ICI response. Screening for *DUX4* tumor expression, with binarized results such as through IHC using anti-DUX4 antibodies, could have clinical utility.

Our data motivate the investigation into *DUX4*'s potential to prognosticate response to ICI. However, our current study is limited by the availability of sufficiently sized ICI-treated cohorts with associated patient data on relevant characteristics such as demographics and risk factors. Additional randomized trial data from diverse metastatic cancer cohorts, with adequate genomic and clinical data, is imperative. As these become available in the future, extending the analyses we have outlined in this study will be important to appraise *DUX4*'s definitive clinical relevance, contextualized among response-modifying clinical variables, in the use of immunotherapy in the treatment of metastatic cancer.

# Materials and methods

## Key resources table

| Reagent type (species) or resource | Designation | Source or reference | Identifiers | Additional information |
|---|---|---|---|---|
| Software, algorithm | UCSC knownGene | *Meyer et al., 2013*; PMID:23155063 | | |
| Software, algorithm | Ensembl 71 | *Flicek et al., 2013*; PMID:23203987 | | |
| Software, algorithm | MISO v2.0 | *Katz et al., 2010*; PMID:21057496 | | |
| Software, algorithm | RSEM v1.2.4 | *Li and Dewey, 2011*; PMID:21816040 | | |
| Software, algorithm | Bowtie v1.0.0 | *Langmead et al., 2009*; PMID:19261174 | | |
| Software, algorithm | TopHat v.2.0.8b | *Trapnell et al., 2009*; PMID:19289445 | | |
| Software, algorithm | Trimmed mean of *M* values (TMM) method | *Robinson and Oshlack, 2010*; PMID:20196867 | | |
| Software, algorithm | clusterProfiler | *Wu et al., 2021*; *Yu et al., 2012*; PMID:34557778, 22455463 | | |
| Software, algorithm | samtools | *Li et al., 2009*; PMID:19505943 | | |
| Software, algorithm | kallisto v.0.46.1 | *Bray et al., 2016*; PMID:27043002 | | |

*Continued on next page*

*Continued*

| Reagent type (species) or resource | Designation | Source or reference | Identifiers | Additional information |
|---|---|---|---|---|
| Software, algorithm | Integrative Genomics Viewer | *Thorvaldsdóttir et al., 2013*; PMID:22517427 | | |
| Software, algorithm | survival | *Therneau and Grambsch, 2000*; *Therneau, 2022*; https://github.com/therneau/survival | | |
| Software, algorithm | stats | *R Development Core Team, 2022*; https://www.r-project.org/ | | |
| Software, algorithm | caret | *Kuhn, 2022*; https://github.com/topepo/caret/ | | |
| Software, algorithm | ggplot2 | *Wickham, 2016*; https://github.com/tidyverse/ggplot2 | | |
| Software, algorithm | dplyr | *Wickham et al., 2022*; https://github.com/tidyverse/dplyr | | |
| Software, algorithm | survminer | *Kassambara et al., 2021*; https://rpkgs.datanovia.com/survminer/index.html | | |
| Software, algorithm | randomForestSRC | *Ishwaran et al., 2008*; https://www.randomforestsrc.org/articles/survival.html | | |
| Software, algorithm | fastshap | *Greenwell, 2021*; https://github.com/bgreenwell/fastshap | | |
| Software, algorithm | pammtools | *Bender and Scheipl, 2018*; https://adibender.github.io/pammtools/ | | |
| Software, algorithm | plotly | *Sievert, 2020*; https://plotly.com/r/ | | |
| Software, algorithm | timeROC | *Blanche et al., 2013*; https://CRAN.R-project.org/package=timeROC | | |
| Software, algorithm | pec | *Mogensen et al., 2012*; https://CRAN.R-project.org/package=pec | | |

## Datasets analyzed in this study

| Dataset name | Accession number(s) | PMID(s) |
|---|---|---|
| 2013/TCGA.ACC | NCI Genomic Data Commons | |
| 2013/TCGA.BLCA | NCI Genomic Data Commons | |
| 2013/TCGA.BRCA | NCI Genomic Data Commons | |
| 2013/TCGA.CESC | NCI Genomic Data Commons | |
| 2013/TCGA.CHOL | NCI Genomic Data Commons | |
| 2013/TCGA.COAD | NCI Genomic Data Commons | |
| 2013/TCGA.DLBC | NCI Genomic Data Commons | |
| 2013/TCGA.ESCA | NCI Genomic Data Commons | |
| 2013/TCGA.GBM | NCI Genomic Data Commons | |
| 2013/TCGA.HNSC | NCI Genomic Data Commons | |
| 2013/TCGA.KICH | NCI Genomic Data Commons | |
| 2013/TCGA.KIRC | NCI Genomic Data Commons | |
| 2013/TCGA.KIRP | NCI Genomic Data Commons | |
| 2013/TCGA.LAML | NCI Genomic Data Commons | |
| 2013/TCGA.LGG | NCI Genomic Data Commons | |
| 2013/TCGA.LIHC | NCI Genomic Data Commons | |
| 2013/TCGA.LUAD | NCI Genomic Data Commons | |
| 2013/TCGA.LUSC | NCI Genomic Data Commons | |

*Continued*

| Dataset name | Accession number(s) | PMID(s) |
|---|---|---|
| 2013/TCGA.MESO | NCI Genomic Data Commons | |
| 2013/TCGA.OV | NCI Genomic Data Commons | |
| 2013/TCGA.PAAD | NCI Genomic Data Commons | |
| 2013/TCGA.PCPG | NCI Genomic Data Commons | |
| 2013/TCGA.PRAD | NCI Genomic Data Commons | |
| 2013/TCGA.READ | NCI Genomic Data Commons | |
| 2013/TCGA.SARC | NCI Genomic Data Commons | |
| 2013/TCGA.SKCM | NCI Genomic Data Commons | |
| 2013/TCGA.STAD | NCI Genomic Data Commons | |
| 2013/TCGA.TGCT | NCI Genomic Data Commons | |
| 2013/TCGA.THCA | NCI Genomic Data Commons | |
| 2013/TCGA.THYM | NCI Genomic Data Commons | |
| 2013/TCGA.UCEC | NCI Genomic Data Commons | |
| 2013/TCGA.UCS | NCI Genomic Data Commons | |
| 2013/TCGA.UVM | NCI Genomic Data Commons | |
| 2014/su2c.prostate_cancer | phs000915.v1.p1 (dbGaP) | 26000489 |
| 2016/chen-chen.acute_lymphoblastic_leukemia | Chinese Genotype-phenotype Archive | 27428428 |
| 2016/fioretos.acute_lymphoblastic_leukemia | EGAD00001002112 (EGA) | 27265895 |
| 2016/mano.acute_lymphoblastic_leukemia | JGAS00000000047 (JGA) | 27019113 |
| 2015/garraway-schadendorf.melanoma_checkpoint_blockade | phs000452.v2.p1 (dbGaP) | 26359337 |
| 2016/hammerbacher.melanoma_checkpoint_blockade | | 27956380 |
| 2016/lo.melanoma_checkpoint_blockade | GSE78220 (GEO) | 26997480 |
| 2017/chinnaiyan.metastatic_cancer | phs000673.v3.p1 (dbGaP) | 28783718 |
| 2017/yang-yeoh.acute_lymphoblastic_leukemia | EGAD00001002151 (EGA) | 27903646 |
| 2018/perou.metastatic_breast_cancer | phs000676.v2.p2 (dbGaP) | 29480819 |
| 2018/powles.urothelial_cancer_checkpoint_blockade | EGAD00001003977 (EGA) | 29443960 |
| 2018/van_allen-choueiri.clear_cell_checkpoint_blockade | phs001493.v1.p1 (dbGaP) | 29301960 |

## Genome annotations, gene expression, and GO enrichment analyses

A genome annotation was created through merging of the UCSC knownGene (*Meyer et al., 2013*), Ensembl 71 (*Flicek et al., 2013*), and MISO v2.0 (*Katz et al., 2010*) annotations for the hg19/GRCh37 assembly. Furthermore, this annotation was expanded by generating all possible combinations of annotated 5′ and 3′ splice sites within each gene. RNA-seq reads were mapped to the transcriptome using RSEM v1.2.4 (*Li and Dewey, 2011*) calling Bowtie v1.0.0 (*Langmead et al., 2009*), with the option '-v 2'. TopHat v.2.0.8b (*Trapnell et al., 2009*) was used to map the unaligned reads to the genome and to the database of splice junctions obtained from the annotation merging described previously. Gene expression estimates (TPM, transcripts per million) obtained were normalized using the trimmed mean of *M* values (TMM) method (*Robinson and Oshlack, 2010*). Endogenous expression of *DUX4* during early embryogenesis range from approximately 2 to 10 TPM (*Chew et al., 2019*; *Hendrickson et al., 2017*). We have therefore defined DUX4-positive samples as those with expression levels >1 TPM. In the differential gene expression analyses for the DUX4-positive vs. -negative comparison, gene expression values per sample group were compared using a two-sided Mann–Whitney *U* test. Differentially expressed genes illustrated in *Figure 2B* were identified as those with an absolute $\log_2$(fold-change) $\geq \log_2$(1.25) and a p-value <0.05. GO enrichment analyses, using the clusterProfiler package (*Wu et al., 2021*; *Yu et al., 2012*), were performed on *DUX4*-upregulated

or -downregulated genes [absolute log$_2$(fold-change) ≥ log$_2$(1.5) and a p-value <0.05] compared against the set of coding genes. Significant GO terms were defined as 'Biological Process' terms with a Benjamini–Hochberg FDR-adjusted p-value <0.05. The top 25 significant GO terms were illustrated (*Figure 2A* and *Figure 2—figure supplement 1A*). To investigate *DUX4* RNA-seq coverage patterns, a fasta file containing the *DUX4* cDNA sequence was assembled, indexed using samtools (*Li et al., 2009*), and used as a reference for read pseudoalignment by kallisto v.0.46.1 (*Bray et al., 2016*). The following kallisto parameters were used: kmer size of 31, estimated fragment length of 200, and estimated fragment length standard deviation of 80. Usage of the single-end option ('--single') and bias correction ('--bias') were also specified. *DUX4* read coverage was visualized using the Integrative Genomics Viewer (IGV, *Thorvaldsdóttir et al., 2013*).

## Gene signature analyses

For a given gene set, *z*-score normalization of the expression values per gene was performed across the patient cohort. The signature score was defined as the mean of the normalized values across the genes of the set.

## Survival analyses, goodness-of-fit measures, and risk modeling

KM estimation, p-value estimates from the log-rank test, and Cox PH regression in the univariate and multivariate contexts were performed using the survival package (*Therneau, 2022*; *Therneau and Grambsch, 2000*). Goodness-of-fit evaluations of the Cox PH models were done by measuring the Akaike information criterion (AIC) and Bayesian information criterion (BIC). AIC and BIC metrics balance model complexity with maximized likelihood, penalizing feature number increases without a concomitant improvement in performance. The likelihood ratio test was also used to compare goodness of fit of full (all variables) vs. reduced (subset of variables) Cox PH models. Specifically, the null hypothesis that the simple model provides as good as a fit as the more complex model was evaluated. The AIC, BIC, and likelihood ratio test p-values were computed using R's stats package (*R Development Core Team, 2022*). For the Cox PH risk modeling, the patients were randomly assigned into training (70%) and test (30%) datasets. The createDataPartition() function from the caret package (*Kuhn, 2022*) was used to preserve the *DUX4* status class distribution after splitting. Full and reduced Cox PH models were created using the training data and the risk scores for each respective model were calculated using caret's predict.coxph() function. For a given patient, the calculated risk score is equal to the HR relative to a 'reference patient' (an individual whose covariate values are set to the respective means, from the training set). Specifically, the risk score is the quotient of the patient's and the reference's exponentiated linear predictors (the sum of the covariates in the model, weighted by the model's regression coefficients). A 'reference risk score' for each model was defined as the median risk score in the training data. Patients were assigned into low- or high-risk groups if their risk scores were lower or higher than the reference, respectively. The trained models were used to calculate risk scores and assign risk labels (based on the training set risk score reference) in the test set. The survival difference between low- and high-risk patients was empirically assessed via KM estimation and the log-rank test. Visualizations were created using the ggplot2 (*Wickham, 2016*), dplyr (*Wickham et al., 2022*), and survminer (*Kassambara et al., 2021*) packages.

## Random Survival Forest, feature importance, and partial dependence

We implemented an RSF model, an ensemble of multiple base learners (survival trees), using the randomForestSRC package (*Ishwaran et al., 2008*). The RSF algorithm is an extension of the Random Forest Algorithm (*Breiman, 2001*) for usage with right-censored data. Here, B bootstrap datasets are created from the original data, used to grow B concomitant survival trees (usually constrained by membership size in the terminal nodes) constructed using a randomly selected subset of the variables. Terminal node statistics are obtained for each tree: the survival function (via the KM estimator), the cumulative hazard function (CHF, via the Nelson–Aalen estimator), and mortality (expected number of deaths; sum of the CHF over time). The RSF prediction is the average across the forest. Of note, each bootstrap dataset excludes 36.8% of the original data on average, the OOB samples. Thus, predictions for a particular sample can be made using the subset of the trees for which it was excluded from training (OOB predictions). Similarly, the associated OOB error for the RSF model can calculated, representing an unbiased estimate of the test error. We

randomly assigned patients into training (70%) and test (30%) datasets. Since the *DUX4*-positive status was a minority class, we utilized the createDataPartition() function from the caret package (*Kuhn, 2022*) to preserve the class distribution within the splits. To determine optimal hyperparameters, we evaluated 5616 RSF models representing different combinations of ntree (number of trees), nodesize (minimum terminal node size), mtry (number of randomly selected splitting variables), na.action (handling of missing data), splitrule (splitting rule), and samptype (type of bootstrap). We selected the model with hyperparameters which minimized both the OOB training and the test errors (defined as 1 − concordance index), namely: ntree = 1500, nodesize = 15, mtry = 3, na.action = "na.impute", splitrule = "bs.gradient", and samptype = "swr". We specified the use of an nsplit (number of random splits) value of 0 to indicate evaluation of all possible split points and usage of the optimum. For test set predictions, patients with missing data were omitted (na.action = "na. omit").

Feature importance in the final RSF model was evaluated using three metrics. First, permutation importance was measured using randomForestSRC's subsample() function. RSF permutation importance utilizes OOB values: a variable's OOB data is permuted and the change in the new vs. original OOB prediction error is quantified. The RSF permutation importance values were standardized by dividing by the variance and multiplying by 100, and the variance and confidence regions were obtained via the delete-*d* jackknife estimator (*Ishwaran and Lu, 2019*). Second, the tree-based feature importance metric minimal depth was calculated using randomForestSRC's var.select() function. The minimal depth threshold (mean minimal depth) is the tree-averaged threshold (conservative = "medium"). Last, Shapley values were estimated using the fastshap package (*Greenwell, 2021*), using 1000 Monte Carlo repetitions. For each prediction, the sum of the estimated Shapley values was corrected (adjust = TRUE) to satisfy the efficiency (or local accuracy) property: for an individual *i*, the sum of *i*'s feature contributions equal the difference between the prediction for *i* and the average prediction across the entire cohort. For the overall measure of importance, the Shapley values were estimated from the mortality predictions from the RSF model (*Figure 5—figure supplement 1E*). Mortality is defined as the number of expected deaths over the observation window. That is, if all patients in the cohort shared the same covariate values as patient *i* who has mortality $m_i$, then an average of *m* deaths is expected (*Ishwaran et al., 2008*). For the time-dependent implementation, we estimated Shapley values associated with the per timepoint RSF survival probability predictions along the observation window (*Figure 5C*).

The relationships of *DUX4* expression and TMB to mortality or survival probability (marginal contributions) were assessed via Shapley dependence plots and partial dependence plots. Partial dependence values were obtained using randomForestSRC's partial() function and OOB predictions for mortality and survival probability were used as input. Visualizations were created in the R programming environment using the dplyr (*Wickham et al., 2022*), ggplot2 (*Wickham, 2016*), pammtools (*Bender and Scheipl, 2018*), and plotly (*Sievert, 2020*) packages.

## Measuring survival model predictive accuracy

The time-dependent ROC curve analyses were done to evaluate the RSF model's accuracy in differentiating patients who die before a particular time *t*, from those who survive past *t* (*Heagerty and Zheng, 2005*). Specifically, for each timepoint, the cumulative/dynamic area under the ROC curve ($AUC^{C/D}$) was calculated by computing the sensitivity (true positive rate) and specificity (1 − false positive rate) associated with using RSF-predicted mortality as the prognostic marker. The time-dependent $AUC^{C/D}$ and 95% confidence interval per timepoint were estimated using the timeROC package, which adds the inverse-probability-of-censoring weights (IPCW) to the sensitivity calculation to correct for selection bias due to right-censoring (*Blanche et al., 2013*). The OOB (training) or the test mortality predictions were used as input. The time-dependent Brier score and the Continuous Ranked Probability Score (CRPS, integrated Brier score divided by time) for the Cox PH models were computed using the pec package (*Mogensen et al., 2012*). The time-dependent Brier score and the CRPS for the RSF model was calculated using the randomForestSRC package (*Ishwaran et al., 2008*). The KM estimator for the censoring times was used to estimate the IPCW (cens.model = "marginal"). Harrell's concordance index for the Cox PH and RSF models was calculated using the survival (*Therneau, 2022*; *Therneau and Grambsch, 2000*) and randomForestSRC packages, respectively. Visualizations were created in the R programming environment using the dplyr and ggplot2 (*Wickham, 2016*) packages.

## Acknowledgements

RKB was supported in part by the NIH/NCI (R01 CA251138), NIH/NHLBI (R01 HL128239, R01 HL151651), and the Blood Cancer Discoveries Grant program through the Leukemia & Lymphoma Society, Mark Foundation for Cancer Research, and Paul G Allen Frontiers Group (8023-20). RKB is a Scholar of The Leukemia & Lymphoma Society (1344-18) and holds the McIlwain Family Endowed Chair in Data Science. The results shown here are in part based upon data generated by the TCGA Research Network: https://cancergenome.nih.gov/.

## Additional information

### Competing interests

Robert K Bradley: This author is an inventor on a patent application submitted by Fred Hutchinson Cancer Center that covers DUX4 expression in cancers and response to immunotherapy (PCT/US2019/043396). This author is a founder and scientific advisor of Codify Therapeutics and Synthesize Bio and holds equity in both companies. This author has received research funding from Codify Therapeutics unrelated to the current work. The other author declares that no competing interests exist.

### Funding

| Funder | Grant reference number | Author |
| --- | --- | --- |
| National Cancer Institute | R01 CA251138 | Robert K Bradley |
| National Heart, Lung, and Blood Institute | R01 HL128239 | Robert K Bradley |
| National Heart, Lung, and Blood Institute | R01 HL151651 | Robert K Bradley |
| The Leukemia & Lymphoma Society | 1344-18 | Robert K Bradley |

The funders had no role in study design, data collection, and interpretation, or the decision to submit the work for publication.

### Author contributions

Jose Mario Bello Pineda, Conceptualization, Formal analysis, Investigation, Visualization, Writing - original draft, Writing – review and editing; Robert K Bradley, Conceptualization, Data curation, Funding acquisition, Writing – review and editing

### Author ORCIDs

Jose Mario Bello Pineda ⓘ http://orcid.org/0000-0003-1417-9200
Robert K Bradley ⓘ https://orcid.org/0000-0002-8046-1063

Reviewer #1 (Public review): https://doi.org/10.7554/eLife.89017.3.sa1
Reviewer #2 (Public review): https://doi.org/10.7554/eLife.89017.3.sa2
Author response https://doi.org/10.7554/eLife.89017.3.sa3

## Additional files

### Supplementary files

• Supplementary file 1. Cox Proportional Hazards regression for overall survival. Hazard ratios, 95% confidence intervals, and p-values for patient covariates in the IMvigor210 urothelial carcinoma cohort, estimated under either the univariate or multivariate Cox Proportional Hazards model. Adjusted p-values were estimated via the Benjamini–Hochberg correction.

• Supplementary file 2. Cox Proportional Hazards regression for overall survival (TGFB1 expression included). As in *Supplementary file 1*, but *TGFB1* expression is included in the models.

• Supplementary file 3. Likelihood ratio test. Estimated p-values from the likelihood ratio test,

comparing the goodness of fit of the specified competing models.

- MDAR checklist

### Data availability

Accession information for RNA sequencing and patient data analyzed in this study are detailed in the appendix. The DUX4 transcript sequence from the hg19 assembly was obtained from the UCSC Genome Browser (https://genome.ucsc.edu/). Any additional information required to reanalyze the data reported in this paper is available from the corresponding author upon request.

The following previously published dataset was used:

| Author(s) | Year | Dataset title | Dataset URL | Database and Identifier |
|---|---|---|---|---|
| Mariathasan S, Turley SJ, Nickles D, Castiglioni A, Yuen K, Wang Y, Kadel III EE, Koeppen H, Astarita JL, Cubas R, Jhunjhunwala S, Banchereau R, Yang Y, Guan Y, Chalouni C, Ziai J, Şenbabaoğlu Y, Santoro S, Sheinson D, Hung J, Giltnane JM, Pierce AA, Mesh K, Lianoglou S, Riegler J, Carano RAD, Eriksson P, Höglund M, Somarriba L, Halligan DL, van der Heijden MS, Loriot Y, Rosenberg JE, Fong L, Mellman I, Chen DS, Green M, Derleth C, Fine GD, Hegde PS, Bourgon R, Powles T | 2018 | Tumor gene expression profiles for the study 'TGF-β attenuates tumour response to PD-L1 blockade by contributing to exclusion of T cells' | https://ega-archive.org/datasets/EGAD00001003977 | European Genome-Phenome Archive, EGAD00001003977 |

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
