## [Editor Report · eLife assessment]

This study presents a **valuable** finding on the association between DUX4 expression with features of immune evasion in human tissue and clinical outcomes in patients with advanced urothelial cancer. The evidence supporting the claims of the authors is **convincing**, using a range of corroborative statistical techniques. Compared to an earlier version, the quality of the manuscript has been enhanced, for example Figure 5 now illustrates the key features of survival probability estimates over time for patients assigned to with the test or training set.

---

## [Referee Report · Reviewer #1 (Public review)]

Pineda et al investigate the association of the hypothesis that Dux4, an embryonic transcription factor, expression in tumor cells is associated with immune evasion and resistance to immunotherapy. They analyze existing cohorts of bulk RNAseq sequenced tumors across cancer types to identify Dux4 expression and association with survival. They find that Dux4 expression is detected in a higher proportion of metastatic tumors compared to primary tumors, is associated with decreased immune infiltrate and a variety of immune metrics and previously nominated immune signatures, and do an in depth evaluation of a cohort of metastatic urothelial cell carcinoma, finding that Dux4 expression is associated with a more immunodeficient tumor microenvironment (desert or excluded microenvironment) and worse survival in this aPDL1 treated cohort. They then find that Dux4 expression is a major independent predictor of survival in this cohort using different types of survival analyses (KM, Cox PH, and random survival forests). With prior existing biological data supporting the hypothesis (in prior work, the senior author has demonstrated Dux4 expression causally suppresses MHC-I expression in interferon-gamma treated cell lines), the current work links Dux4 expression with less immune activity in clinical tumor samples and with survival in ICI treated urothelial carcinomas, and demonstrates that Dux4 expression provides independent information towards survival including other molecular and clinical characteristics (TMB, ECOG PS as the other strongest markers), and provides interesting resolution on landmark analyses with TMB and Dux4 expression providing greater informativeness at later survival landmarks (e.g. 1 year and later), while ECOG PS has strong informativeness already at earlier time points. This work provides impetus towards more mechanistic and functional dissection of the mechanism of Dux4-associated changes with the tumor microenvironment (e.g. in vivo mouse studies) as well as potential interventional studies (e.g. Dux4 as a target in combination therapies). What the work does not provide is additional resolution on the mechanism of how Dux4 may be associated with a more immunodeficient microenvironment.

The conclusions are generally well supported, but there are issues that would benefit from clarification and extension:

- The finding that Dux4 expression is detected in a higher proportion of metastatic tumors and at higher levels compared to TCGA samples (Fig 1BC) is striking. However, at least for one tumor type (melanoma), the TCGA cohort is comprised of mostly locoregional metastatic (n=81 primary and 367 metastatic tumors in the PanCan Atlas). Since there are annotations for primary and (locoregional) metastatic samples in TCGA, an analysis of the primary vs. locoregional metastasis vs distant metastatic samples seems reasonable and likely informative. The analysis of tumors with matched FFPE and flash frozen samples with hybrid probe capture and polyA sequencing, respectively is a nice validation to show that the difference in Dux4 expression is not due to differences in preservation of starting material/sequencing in the metastatic samples vs TCGA samples (S1BC).

- The findings that Dux4 expression in the metastatic urothelial carcinoma setting is associated with a more immunodeficient microenvironment (Figure 2) is clear and unambiguous using multiple lines of data and analyses (bulk RNAseq, DUX4-positive vs DUX4-negative tumors, different immune cell and cytokine signatures; IHC showing an association with immune deserts and immune excluded phenotypes). However, this is an association and does not demonstrate causality.

- The survival analyses (Fig 3,4,5) show fairly convincingly that Dux4 provide independent predictive information beyond clinical variables and TMB towards survival in the aPDL1 treated metastatic urothelial carcinoma cohort. However, the choice to split the cohort into Dux4 negative (defined as < 0.25 TPM) and Dux4 positive (> 1 TPM) while excluding a large number of patients (n=126 pts) that fall in between has significant impact on the rigor of conclusions. This would benefit from showing all the data (e.g. including the 3rd group of in-betweens in the survival analyses as a separate group).

- The authors demonstrate that adding Dux4 to clinical markers and TMB results in an improved predictive model for survival, but there are a few questions regarding this model as a clinical biomarker

o Is Dux4 expression better than other correlated immune signatures/markers (e.g. interferon gamma, T effector signature, overall immune infiltrate) in providing additional information?

- The use of random survival forests to quantify the (predictive) marginal effect of Dux4+ vs Dux4- expression on survival in a non-parametric model as well as shed light on association with survival at different landmark times using Shapley values is quite interesting and well conducted.

---

## [Referee Report · Reviewer #2 (Public review)]

Summary:

This article takes an expansive look at the potential role of DUX4 in cancer treatment and prognosis, including its correlation with other key biomarkers, the potential for cancer to be resistant to treatment, and risk prediction.

Strengths:

The primary strength of this work is the breadth of the analyses. The authors have linked DUX4 to not just one but multiple points in the trajectory of cancer, which increases the face validity of their conclusion that DUX4 is meaningfully related to the course of a cancer as well as the prognosis for a patient.

Statistically, the authors have taken care to properly validate their findings using appropriate bootstrapping and testing strategies.

Weaknesses:

Several weaknesses are noted. First, there is little-to-no description of the underlying sample population. It is only stated that "several large cohorts of patients with different metastatic cancers" were analyzed, and that a cohort of patients with advanced urothelial cancer was used for estimating associations with clinical outcomes. Lacking is information on the sampling mechanism, inclusion/exclusion criteria, treatment modalities, the definition of 'time = 0', the number of events observed, or even the sample size. Knowledge about the underlying study design would help explain some counterintuitive results, e.g. that the hazard of death among patients with Stage IV cancer is half that of those with Stage I cancer (Table 1); presumably this is not because Stage IV is actually protective but rather an artifact of the sampling scheme for these data. Second, the definition of negative versus positive DUX4 expression varies throughout the paper. In Figure 2B, Figure 3A, and Figure 3C, it is defined as >1 TPM vs. <= 1 TPM; in Figure 4A and Figure 5A, it is defined as >1 TPM vs. < 0.25 TPM; in Figure S1C it is partitioned into four groups, with boundaries defined at 0.25 TPM, 1 TPM, and 5 TPM. If categorization is needed, a rationale should be provided (ideally prospectively and not based upon the observed data, so as to avoid the perception of forking paths analyses), and it should be consistently applied. Third and finally, data seem to be occasionally excluded without rationale. For example, as mentioned above, the Cox model presented in Figure 4A seems to exclude all patients with DUX4 TPM between 0.25 and 1. Figure 3C excludes patients with TMB in the lowest quartile (although the decision was ostensibly to control for TMB confounding, there are more appropriate ways to do so that don't result in loss of data, e.g. a stratified KM plot). Excluding patients based upon a particular region of the covariate space makes interpreting the resulting model awkward.

---

## [Author Response]

The following is the authors’ response to the original reviews.

**Reviewer #1:**
Figure 1The "matched primary tumors" from TCGA include n=424 from cutaneous melanoma; but it is unclear where this is coming from; the PanCan Atlas for melanoma shows n=81 primary and 367 metastatic tumors. There are also additional large cohorts of ICI-treated metastatic tumors with RNAseq data (e.g. a metastatic melanoma cohort with 100+ patients https://doi.org/10.1038/s41591-019-0654-5) that would increase the numbers here.

We thank the reviewer for their observation. We have replaced references to “primary” cancers as “TCGA” cancers as appropriate. While the TCGA analyses included metastatic samples, the majority of the TCGA tumors in most cohorts correspond to primary cancers or local metastases, a point which we added to the text. We retained Fig. 1D as the representative examples are actual primary samples. We have decided to defer analysis of additional melanoma cohorts for future inquiry.

Figure 2What is the basis for the split between high and low Dux4 expressing tumors at 1 TPM? Is it arbitrary, or based on some structure in the distribution? (e.g. bimodal distribution)

Our previous analyses of RNA-seq datasets derived from early embryogenesis samples (PMID: 3132774, 28459457) showed that physiologic levels of DUX4 range from approximately 2 to 10 TPM. We added a description in the methods section, under “Genome annotations, gene expression, and Gene Ontology (GO) enrichment analyses,” of our conservative choice for the threshold: DUX4-positivity defined as expression levels > 1 TPM.

Figure 3Overall claim is that Dux4 expression is associated with worse survival in metastatic urothelial carcinomas treated with PD-L1 inhibitor. However, the rationale for the choice of split (Dux4 expression < 0.5 and > 1 TPM) to show is unclear (is this the 25th percentile? 75th percentiles?), and the rationale/interpretation of the "partial adjustment" for TMB by removing the bottom quartile of TMB feels non-rigorous and prone to bias. It doesn't feel like Fig 3bc contributes very much; Figure 4 really is the more rigorous analysis.

We thank the reviewer for these comments and suggestions. We adjusted the analyses in Fig. 3C and Fig. S3 to be consistent with Fig. 1 and Fig. 2, in terms of the choice of split. We also clarified in the text how our initial, crude TMB adjustment served as an important indication for us to pursue more rigorous statistical approaches.

Figure 4Dux4 expression is independently associated with worse survival considering other clinical and molecular characteristicsI would include TGFB in the features considered in the table (in the supplementary but not the main table or forest plots, not sure why not?)The choice of Dux4 expression split ( < 0.25 and > 1 TPM) feels arbitrary and is different than the split in Figure 3; what is the rationale for this? Also, how many patients does this exclude? (TPM between 0.25 and 1). What does the continuous value or median split for Dux4 expression give you for the CoxPH model?Re: building a predictive model, excluding patients (e.g. between <0.25 and > 1 Dux4 TPM) makes the model difficult to apply (e.g. cannot apply to patients with Dux4 levels in the missing interval); a better predictive model would include all patients in the cohort.

We thank the reviewer for their other suggestions. We have clarified in the text that our choice to define DUX4negative samples as those with DUX4 expression levels < 0.25 TPM was made to preemptively address potential misclassifications due to decreased sensitivity of bulk RNA-seq at very low expression levels (PMID: 18516045). We believe our classifications with the new scheme are more reliable. We have also now specified in the text that our categorization excludes 126 patients. We have decided to not pursue the addition of TGFB or exploration of the use of an alternative split or continuous version of DUX4 expression in the Cox Proportional Hazards analyses but appreciate the suggestions, which we will keep in mind for future studies.

Figure 5An RSF (randomized survival forest) model predicts survival in Dux4+ vs Dux4- patient, and the Shapley values for landmark time analyses show time-varying effects of different features.In some sense, the authors have already demonstrated that Dux4+ is associated with survival differences in ICI treated patients; so a model that predicts survival applied to Dux4+ and Dux4- patients that shows a difference in survival is unsurprising (even in a training/test set setting given that there is a difference in survival across the entire cohort). The quantified marginal effect (from a predictive perspective) of different features is what is interesting here. In that light, I'd like to see more validation of the model up front, specifically how close the predicted survival is to the actual survival of patients (e.g. the survival curves in Fig 5a but with actual survival of the Dux4- and Dux4+ cohorts superimposed on the predicted probabilities).

We thank the reviewer for this suggestion. We have added a plot showing the superimposed survival probability estimates over time for the RSF and KM models for patients assigned to either the test or training sets in Fig. 5.

SFig 5Unclear how the authors got estimates of the # of expected deaths associated with covariates (e.g. "...we measured an increase in the number of predicted deaths associated with DUX4-positivity by approximately 16, over DUX4negative status (Fig S5F-G).") from Shapley values as shown in the indicated figure - is this 16 out of the entire cohort? At a given time point? Would recommend perhaps showing the inferred absolute change in mortality (e.g. 8% absolute increase in mortality)

Mortality is the expected number of deaths for the cohort over the observation window, measured as the sum of the CHF over time. We have clarified this in the Methods section, under “Random Survival Forest, feature importance, and partial dependence.” We have also changed the quantification to show the absolute mortality differences comparing patients with DUX4-negative and -positive tumors; we thank the reviewer for this suggestion. We have also clarified in the text that adjusted mortality was estimated via partial dependence, which operates using the correct units, as opposed to Shapley values, where attribution is scaled. Finally, we changed the referenced figure when discussing changes in mortality associated with TMB and DUX4 status (Fig. S5H-I); we appreciate the reviewer pointing out this error.

Figure S1B-CThe authors argue that Dux4 expression is not an artifact of FFPE tissue by analyzing a mixed tumor cohort sequenced with both poly-A and hybrid probe capture in matched flash-frozen and FFPE tumor samples, showing that it is (1) detectible both FFPE and flash-frozen tissue and (2) higher levels are detected in polyA sequencing/frozen tissue. However, the reference for this section (D. Robinson et al 2015) is a study of a cohort of prostate cancers with polyA bulk RNAseq sequencing; is this correct/is the data coming from a different study?Analysis of scRNAseq (if available) would strengthen their analyses by better delineating the expression and response of interferon-gamma and downstream (e.g. antigen presentation) pathways in specific cell compartments, and potential differences in cell-cell interactions (e.g. using CellPhoneDB) associated with Dux4+ vs Dux4- tumors.Do the investigators find similar findings in primary and metastatic tumors sequenced the same way (e.g. tcga primary vs met melanoma, albeit most of the met melanoma are Stage III lymph nodes)?

We thank the reviewer for finding the citation error. We have corrected the manuscript to reflect the correct study we analyzed (PMID: 28783718). We also thank the reviewer for their additional suggestions, which undoubtedly would strengthen the current study. However, we have respectfully decided to defer these additional analyses for future study.

**Reviewer #2:**
It is strange as a statistician to see BIC and AIC represented as barplots, e.g. Figure 4B. There is no knowledge to be gained through this visual representation that would not otherwise be conveyed by just giving the numbers.

We thank the reviewer for this suggestion. We understand that simply stating the numbers would be equally informative. However, we respectfully decided to retain our current versions of Figures 4 and S4 so that the numbers can be illustrated in a visual manner in the figures, rather than just stated in the text.